Three new karst-dwelling Cnemaspis Strauch, 1887 (Squamata; Gekkoniade) from Peninsular Thailand and the phylogenetic placement of C. punctatonuchalis and C. vandeventeri

Wood Jr Perry Lee pwood@byu.edu 1
Grismer L. Lee 2
Aowphol Anchalee 3
Aguilar César A. 1
Cota Micheal 4 5
Grismer Marta S. 2
Murdoch Matthew L. 2
Sites Jr Jack W. 1
1 Department of Biology and Bean Life Science Museum, Brigham Young University , Provo , UT , United States
2 Department of Biology, La Sierra University , Riverside , CA , United States
3 Faculty of Science, Department of Zoology, Kasetsart University , Chatuchak, Bangkok , Thailand
4 Natural History Museum, National Science Museum, Thailand , Technopolis, Khlong 5, Khlong Luang, Pathum Thani , Thailand
5 Suan Sunandha Rajabhat University, Institute for Research and Development , Dusit , Bangkok , Thailand
Crandall Keith
Electronic publication date: 2017 Jan 24
Publication date: 2017
Volume: 5
Electronic Location ID: e2884
Received 2016 Sep 7; Accepted 2016 Dec 8
Copyright: ©2017 Wood Jr et al.
Copyright year: 2017
Copyright holder: Wood Jr et al.
License: This is an open access article distributed under the terms of the Creative Commons Attribution License, which permits unrestricted use, distribution, reproduction and adaptation in any medium and for any purpose provided that it is properly attributed. For attribution, the original author(s), title, publication source (PeerJ) and either DOI or URL of the article must be cited.
License URL: https://creativecommons.org/licenses/by/4.0/

Keywords: Limestone forests, chanthaburiensis group, siamensis group, Malay peninsula

Funding: Department of Biology at Brigham Young University NSF dimensions EF-1241885 Doctoral Dissertation Improvement DDIG #1501198 Enhanced Engagement in Research (PEER) Science program PGA-2000003545 US Agency for International Development (USAID) National Science Foundation (NSF) Funding was from the Department of Biology at Brigham Young University. Additional funding for this research is from the NSF dimensions grant EF-1241885 issued to JWS and the Doctoral Dissertation Improvement Grant (DDIG #1501198) issued to PLWJ and JWS. Partnerships for Enhanced Engagement in Research (PEER) Science program (grant PGA-2000003545), which is a partnership between the US Agency for International Development (USAID) and the National Science Foundation (NSF), provided funding for AA. The funders had no role in study design, data collection and analysis, decision to publish, or preparation of the manuscript.

==============================
Three new species of Rock Geckos Cnemaspis lineogularis sp. nov., C. phangngaensis sp. nov., and C. thachanaensis sp. nov. of the chanthaburiensis and siamensis groups are described from the Thai portion of the Thai-Malay Peninsula. These new species are distinguished from all other species in their two respective groups based on a unique combination of morphological characteristics, which is further supported by mitochondrial DNA (mtDNA) from the NADH dehydrogenase subunit 2 gene (ND2). Cnemaspis lineogularis sp. nov. is differentiated from all other species in the chanthaburiensis group by having a smaller maximum SVL 38 mm, 13 paravertebral tubercles, enlarged femoral scales, no caudal bands, and a 19.5–23.0% pairwise sequence divergence (ND2). Cnemaspis phangngaensis sp. nov. is differentiated from all other species in the siamensis group by having the unique combination of 10 infralabial scales, four continuous pore-bearing precloacal scales, paravertebral tubercles linearly arranged, lacking tubercles on the lower flanks, having ventrolateral caudal tubercles anteriorly present, caudal tubercles restricted to a single paraveterbral row on each side, a single median row of keeled subcaudals, and a 8.8–25.2% pairwise sequence divergence (ND2). Cnemaspis thachanaensis sp. nov. is distinguished from all other species in the siamensis group by having 10 or 11 supralabial scales 9–11 infralabial scales, paravertebral tubercles linearly arranged, ventrolateral caudal tubercles anteriorly, caudal tubercles restricted to a single paravertebral row on each side, a single median row of keeled subcaudal scales, lacking a single enlarged subcaudal scale row, lacking postcloaclal tubercles in males, the presence of an enlarged submetatarsal scale at the base if the 1st toe, and a 13.4–28.8% pairwise sequence divergence (ND2). The new phylogenetic analyses place C. punctatonuchalis and C. vandeventeri in the siamensis group with C. punctatonuchalis as the sister species to C. huaseesom and C. vandeventeri as the sister species to C. siamensis, corroborating previous hypotheses based on morphology. The discovery of three new karst-dwelling endemics brings the total number of nominal Thai Cnemaspis species to 15 and underscores the need for continued field research in poorly known areas of the Thai-Malay Peninsula, especially those that are threatened and often overlooked as biodiversity hot spots.

Introduction

The Thai-Malay Peninsula is a long (1,127 km) and narrow (maximum width 322 km) appendix of mainland Asia extending from Indochina in the north to its southern terminus in Singapore. The Thai-Malay Peninsula is comprised of the southern portion of Myanmar, the southwestern section of Thailand, West Malaysia, and Singapore. This region is both geologically and climatically complex and has been influenced by a number of factors. The environmental complexity of this region has helped to form two prominent biogeographic barriers, the Isthmus of Kra and the Kangar-pattani line. These biogeographic barriers serve as pivotal crossroads for faunal exchange between the Indochinese and Sundaic biota (e.g., Raes et al., 2014; De Bruyn et al., 2013; Parnell, 2013; Patou et al., 2009; Woodruff & Turner, 2009; Gorog, Sinaga & Engstrom, 2004; Pauwels et al., 2003; Hughes, Round & Woodruff, 2003; Woodruff, 2003; Grismer et al., 2014d; Grismer, 2011). One feature that is often over-looked in terms of biodiversity are the myriad of limestone forests and karst formations dispersed throughout the Malay Peninsula.

Karstic regions have been referred to as “arks” or biodiversity reservoirs that can be used as stock for repopulating degraded environments during ecosystem reassembly (Schilthuizen, 2004). In addition to serving as arks, karst formations have been known to provide natural laboratories for biogeographic, evolutionary, ecological, and taxonomic research (e.g., Ng, 1991; Grismer et al., 2014c; Grismer et al., 2014b; Schilthuizen et al., 2005; Schilthuizen et al., 1999; Kiew, 1991). From chemical and mechanical weathering karst formations have been molded into a unique suite of microhabitats in which a number of species have become adapted (e.g., Vermeulen & Whitten, 1999; Komo, 1998a; Komo, 1998b; Tija, 1998). To date there has been a fair amount of research conducted on the flora of karst formations and their surrounding limestone forests, resulting in a high estimate of endemic species (Kiew, 1998; Clements et al., 2006; Chin, 1977 and references therein). In addition to the high level of floral endemism there are also high levels of invertebrate endemism associated with karst formations (e.g., Holloway, 1986; Vermeulen & Whitten, 1999). Although these areas harbor a high degree of endemism for invertebrates and plant species they are generally not considered to hold large numbers of endemic terrestrial vertebrates (i.e., Jenkins et al., 2005; Alström et al., 2010; Woxvold, Duckworth & Timmins, 2009), because most vertebrates have high dispersal capabilities. There are only a few mammals and birds that are thought to be restricted to karst formations (e.g., Latinne et al., 2011; Clements et al., 2006). In contrast, recent taxonomic work in Peninsular Malaysia has uncovered an impressive amount of new microendemic karst-dwelling species of reptiles, including 14 new lizards (Grismer et al., 2008a; Grismer et al., 2008b; Grismer et al., 2009; Grismer et al., 2013b; Grismer et al., 2012; Grismer et al., 2014e; Grismer et al., 2014c; Grismer et al., 2013a; Grismer et al., 2016a; Wood Jr et al., 2013) and two new snakes (Grismer et al., 2014b; E Quah, 2017, unpublished data). However, these surveys only covered a small portion of the limestone forests and karst formations of Peninsular Malaysia continue northward up the entire Thai-Malay Peninsula into central Thailand and eastern Myanmar.

Dispersed throughout Peninsular Thailand are hundreds of unexplored isolated karst formations. From the limited number of surveys that have been conducted, a few gekkotan species have been identified and described (e.g., Grismer et al., 2012; Grismer et al., 2015; Pauwels et al., 2013; Grismer et al., 2012; Ellis & Pauwels, 2012). A major focus of these surveys in the last two decades has lead to the discovery of at least 15 new micro-endemic karst-dwelling Bent-toed Geckos in the genus Cyrtodactylus (e.g., Ellis & Pauwels, 2012; Pauwels et al., 2013; Grismer et al., 2016a; Sacha, 2015, and references therein). The genus Cyrtodactylus is a monophyletic assemblage of 224 species (Uetz, Freed & Hošek, 2016) that have a broad range throughout Asia to the western Pacific (Wood Jr et al., 2012). This diverse group of nocturnal geckos occupies a broad suite of microhabitats (eg. karst and granite caves, granite boulders, leaf-litter, limestone forests etc.) with a number of specialists in distantly related clades. One group of convergent specialists that has recently received attention are the karst and cave dwelling ecomoprhs in the condorensis and intermedius species complexes (Grismer et al., 2015, and references therein). These granite and karst-dwelling Cyrtodactylus are often found syntopically with the Rock Geckos in genus Cnemaspis and can pose as a potential competitor.

The Rock geckos of the genus Cnemaspis comprise a clade of 55 described species that are widespread throughout the Sunda Shelf, with a majority of the species being from the Thai-Malay Peninsula and their adjacent islands. Most Cnemaspis are diurnal, cryptically colored, scansorial species, however some species such as C. psychedelica Grismer, Ngo & Grismer 2010 (Fig. 8 in Grismer et al., 2014d) are brightly colored and a number of species dispersed throughout the phylogenetic tree are nocturnal with multiple independent transitions (Fig. 5 in Grismer et al., 2014d). Like Cyrtodactylus, Cnemaspis are also found on a variety of substrates (e.g., granite, karst, vegetation, terrestrial and various combinations of these). Often when there are micro-endemic karst-dwelling Cyrtodactylus (nocturnal) there is usually an endemic diurnal karst-dwelling Cnemaspis occupying the same niche during different activity periods (e.g., Grismer et al., 2016a; Grismer et al., 2012; Grismer et al., 2014e; Grismer et al., 2016b). From this arises a number of interesting questions about niche partitioning, behavior, and potential competition of Cyrtodactylus and Cnemaspis, however this is not the focus of this paper. Recent surveys in Phangnga, Tha Chana, Thap Sakae, Prachuap Khiri Khan, and Tham Sonk hill during the month of September 2016 resulted in the collection of Cnemaspis punctatonuchalis, C. vandeventeri and three undescribed species of Cnemaspis that co-occur with some of the aforementioned species of karst endemic Cyrtodactlyus. These new populations can be placed in the genus Cnemaspis based on having broad, flattened heads; large somewhat forward and upwardly directed eyes with round pupils and no eyelids; flattened bodies; long, widely splayed limbs with long, inflected digits; and no femoral pores. Here we present morphological and color pattern data as evidence, for delimitation of these three new species of Cnemaspis, bolstered by mtDNA genetic data, and present the phylogenetic placement of C. punctatonuchalis and C. vandeventeri.

Materials and Methods

Taxon sampling

We obtained 203 samples of Cnemaspis and outgroups from Grismer et al. (2014d). In combination with this dataset we added 14 new samples including two species of Thai Cnemaspis (C. punctatonuchalis and C. vandeventeri) that have never been sequenced, along with three undescribed species of Cnemaspis from peninsular Thailand (Fig. 1, Table S1). Brigham Young University’s Institutional Animal Care and Use Committee (IACUC) has approved the animal use protocol for this study (protocol # 160401). The electronic version of this article in Portable Document Format (PDF) will represent a published work according to the International Commission on Zoological Nomenclature (ICZN), and hence the new names contained in the electronic version are effectively published under that Code from the electronic edition alone. This published work and the nomenclatural acts it contains have been registered in ZooBank, the online registration system for the ICZN. The ZooBank LSIDs (Life Science Identifiers) can be resolved and the associated information viewed through any standard web browser by appending the LSID to the prefix http://zoobank.org/. The LSID for this publication is: urn:lsid:zoobank.org:pub:987831FC-F4BA-4409-A43C-9929F913E9F9. The online version of this work is archived and available from the following digital repositorie: PubMed Central.

Figure 1 Distribution of the species of Cnemaspis in the chanthaburiensis and siamensis groups.

Stars indicate type localities, colored dots represent additional localities for the respective species, and the colored outlines correspond to colored clades in Fig. 2. The asterisk (*) identifies a species not included in the molecular analysis and is hypothesized to be in the siamensis group based on its distribution (Grismer et al., 2014d).

Molecular and phylogenetic analyses

Genomic DNA was isolated from liver or muscle tissues stored in 95% ethanol using the animal tissue protocol in the Qiagen DNeasy™ tissue kit (Valencia, CA, USA). The mitochondrial gene NADH dehydrogenase subunit 2 (ND2) and the flanking tRNAs (∼1,335 bp) was amplified using a double-stranded Polymerase Chain Reaction (PCR) under the following conditions: 1.0 µl (∼10–33 µg) genomic DNA, 1.0 µl (10 µM) forward primer L4437b (5′-AAGCAGTTGGGCCCATACC-3′), 1.0 µl (10 µM) reverse primer H5934 (5′-AGRGTGCCAATGTCTTTGTGRTT-3′), 1.0 µl deoxynucleotide pairs (1.5 µM), 2.0 µl 5x buffer (1.5 µM), 1.0 MgCl 10x buffer (1.5 µM), 0.18 µl Promega Taq polymerase (5 u/µl), and 7.5 µl H2O, primers are from Macey et al. (1997). All PCR reactions were executed in an Eppendorf Mastercycler gradient theromocycler under the following conditions: initial denaturation at 95 °C for 2 min, followed by a second denaturation at 95 °C for 35 s, annealing at 52 °C for 35 s, followed by a cycle extension at 72 °C for 35s, for 33 cycles. All PCR products were visualized on a 1% agarose gel electrophoresis. Successful targeted PCR products were vacuum purified using MANU 30 PCR Millipore plates and purified products were resuspended in DNA grade water. Purified PCR products were sequenced using the PCR primers from above and sequencing primers CyrtintF1 (5′-TAGCCYTCTCYTCYATYGCCC-3′) and CyrtintR1 (5′-ATTGTKAGDGTRGCYAGGSTKGG-3′) from (Siler et al., 2010) on the ABI Big-Dye Terminator v3.1 Cycle Sequencing Kit in an ABI GeneAmp PCR 9700 thermal cycler. Cycle sequencing reactions were purified with Sephadex G-50 Fine (GE Healthcare) and sequenced on an ABI 3730xl DNA Analyzer at the BYU DNA Sequencing Center (DNASC). All new sequences produced from this study are deposited in GenBank under the following accession numbers KY091231 –KY091244 (Table S1). All sequences were edited and aligned in Geneious v6.1.8 (Kearse et al., 2012), alignment was constructed using the Muscle plugin (Edgar, 2004). Mesquite v3.02 (Maddison & Maddison, 2015) was used to check for stop codons and to ensure the correct amino acid read frame.

Table 1 Models of molecular evolution used for the ML and BI analyses.

Gene	Model selected	Model applied for ML	Model applied for BI	
ND2				
1st pos	GTR + I + Γ	GTR + I + Γ	GTR + I + Γ	
2nd pos	GTR + I + Γ	GTR + I + Γ	GTR + I + Γ	
3rd pos	GTR + I + Γ	GTR + I + Γ	GTR + I + Γ	
tRNAs	TrN + I + Γ	GTR + I + Γ	HKY  + I + Γ	

For estimating the phylogenetic relationships we used both partitioned Maximum Likelihood (ML) and partitioned Bayesian Inference (BI) methods. The ND2 gene was partitioned by codon position and the tRNAs were treated as a single partition for both the ML and BI analyses. All models of molecular evolution were estimated in ModelTest v3.7 (Posada & Crandall, 1998), using the Bayesian Information Criterion (BIC). The best fit models of evolution are in presented in Table 1. The partitioned ML analyses was performed using RAxML HPC v7.5.4 (Stamatakis, 2006), 1,000 bootstrap pseudoreplicates via therapid hill-climbing algorithm (Stamatakis, Hoover & Rougemont, 2008) with 200 searches for the best tree. The Bayesian analysis was carried out in MrBayes v3.2 (Huelsenbeck et al., 2001; Ronquist et al., 2012) using the default priors. Two simultaneous runs were performed with eight chains per run, seven hot and one cold following default priors. The analysis was run for 2 x 106 generations and sampled every 1000 generations from the Markov Chain Monte Carlo (MCMC). The analysis was halted after the average standard deviation split frequency was below 0.01 and we assumed convergence. We conservatively discarded the first 25% of the trees as burnin and constructed a consensus tree using the sumt command in MrBayes. Nodes having bootstrap support values (BS) greater than 70 and posterior probabilities (PP) above 0.95 were considered well supported (Huelsenbeck et al., 2001; Wilcox et al., 2002). We calculated uncorrected percent sequence divergences for ND2 in Mega v6.06 (Tamura et al., 2013).

Morphological analyses

Morphological and color pattern characteristics follow Grismer et al. (2014d): color pattern characters were taken from digital images of living specimens cataloged in the La Sierra University Digital Photo Collection (LSUDPC) and from living specimens in the field. The following measurements on the type series were taken by PLWJ with a electronic digital caliper to the nearest 0.1 mm, under a Lica WILD M10 dissecting microscope on the left side of the body where appropriate: snout-vent length (SVL), taken from the tip of snout to the vent; tail length (TL), taken from the vent to the tip of the tail, original or regenerated; tail width (TW), taken at the base of the tail immediately posterior to the postcloacal swelling; forearm length (FL), taken on the dorsal surface from the posterior margin of the elbow while flexed 90° to the inflection of the flexed wrist; tibia length (TBL), taken on the ventral surface from the posterior surface of the knee while flexed 90° to the base of the heel; axilla to groin length (AG), taken from the posterior margin of the forelimb at its insertion point on the body to the anterior margin of the hind limb at its insertion point on the body; head length (HL), the distance from the posterior margin of the retroarticular process of the lower jaw to the tip of the snout; head width (HW), measured at the angle of the jaws; head depth (HD), the maximum height of head from the occiput to the throat; eye diameter (ED), the greatest horizontal diameter of the eyeball; eye to ear distance (EE), measured from the anterior edge of the ear opening to the posterior edge of the eyeball; eye to snout distance (ES), measured from anteriormost margin of the eyeball to the tip of snout; eye to nostril distance (EN), measured from the anteriormost margin of the eyeball to the posterior margin of the external nares; inner orbital distance (IO), the width of the frontal bone at the level of the anterior edges of the orbit; ear length (EL), the greatest vertical distance of the ear opening; and internarial distance (IN), measured between the medial margins of the nares across the rostrum. Additional character states evaluated were numbers of supralabial and infralabial scales counted from below the middle of the orbit to the rostral and mental scales, respectively; the texture of the scales on the anterior margin of the forearm; the number of paravertebral tubercles between limb insertions counted in a straight line immediately left of the vertebral column (where applicable); the presence or absence of a row of enlarged, widely spaced, tubercles along the ventrolateral edge of the body (flank) between the limb insertions; the general size (i.e., strong, moderate, weak) and arrangement (i.e., random or linear) of the dorsal body tubercles; the number of subdigital lamellae beneath the fourth toe counted from the base of the first phalanx to the claw; the distribution of transverse and granular subdigital lamellae on the fourth toe; the total number of precloacal pores, their orientation and shape; the number of precloacal scales lacking pores separating the left and right series of pore-bearing precloacal scales; the degree and arrangement of body and tail tuberculation; the relative size and morphology of the subcaudal scales, subtibial scales, and submetatarsal scales beneath the first metatarsal; and the number of postcloacal tubercles on each side of the tail base. Longitudinal rows of caudal tubercles on the non-regenerated portion of the tail are quite variable between species and useful in differentiating several taxa. Up to five pairs of the following rows may be present in varying combinations: paravertebral row—the dorsal row adjacent to the middorsal, caudal furrow; dorsolateral row—the row between the paravertebral row and the lateral, caudal furrow on the dorsolateral margin of the tail; lateral row—the row immediately below the lateral, caudal furrow; and ventrolateral row—the row below the lateral row on the ventrolateral margin of the tail below the lateral caudal furrow. When present, this row is usually restricted to the anterior 25% (or less) of the tail. Occasionally there may be a row of tubercles within the lateral, caudal furrow.

Figure 2 Phylogenetic relationships of the chanthaburiensis (A) and the siamensis (B) groups.

Right, Maximum Likelihood tree from RAxML (−ln L 60818.390304) for all species of Cnemaspis and outgroups with bootstrap support values (BS) and Bayesian posterior probabilities (PP), respectively. Country abbreviations for the tip labels are as follows: CM, Cambodia; TH, Thailand; WM, West Malaysia; VT, Vietnam. All new species are highlighted with grey boxes and the genus Cnemaspis is highlighted using a box with dashed lines in the main tree.

Results

The phylogenetic analyses place both C. punctatonuchalis and C. vandeventeri in the siamensis group (Fig. 2). Cnemaspis punctatonuchalis is strongly recovered for the ML analysis (100 BS) but not the BI (0.87 PP) as the sister species to C. huaseesom. Cnemaspis vandeventeri is strongly supported (100 BS and 0.99 PP) as the sister lineage to C. siamensis. Phylogenetic analyses of the three new populations sampled from Prachuap Khiri Khan, Phangnga, and Tha Chana represent well-supported independent lineages (100 BS, 1.0 PP; 100 BS, 1.0 PP; 100 BS, 1.0 PP, respectively). The samples from Wat Khao Daeng are well-supported (100 BS, 1.0 PP) as the sister lineage to the chanthaburiensis group (Fig. 2A) and demonstrate a 19.5–23% mtDNA pairwise sequence divergence from all of the other species in this group (Table 2). Both the Phangnga and the Tha Chana populations are nested within the siamensis group (Fig. 2B). The Phangnga population is well-supported for ML (99 BS) but lacks support from the BI (0.56 PP) as the sister lineage to a clade composed of C. omari and C. roticani and demonstrate a 8.8–25.2% mtDNA pairwise sequence divergence from all of the other species in the siamensis group (Table 3). The population from Tha Chana forms a well-support lineage (100 BS and 1.0 PP) and is strongly (100 BS and 1.0 PP) placed as the sister lineage to a clade composed of C. siamensis and C. vandeventeri and bares a 13.4–28% mtDNA pairwise sequence divergence form all of the other species in the siamensis group (Table 3). Given that these new populations form well-supported independent lineages (Figs. 2A and 2B) coupled with high genetic distances and a unique set of morphological and color pattern characteristics that separate them from all members of their respective groups, we describe these three populations below as new species.

Table 2 Pairwise uncorrected p-distances based on 1,335 bp of ND2 and associated tRNAs calculated in MEGA v6.06 (Tamura et al., 2013) within the chanthaburiensis group.

Within species distances are presented in bold text and between species distances are presented below the diagonal.

	C. lineogularis sp. nov.	C. aurantiacopes	C. caudanivea	C. chanthaburiensis	C. neangthyi	C. nuicamensis	C. tucdupensis	
C. lineogularis sp. nov.	0.003							
C. aurantiacopes	0.200	0.011						
C. caudanivea	0.195	0.136	0.002					
C. chanthaburiensis	0.217	0.161	0.167	–				
C. neangthyi	0.230	0.161	0.169	0.156	0.001			
C. nuicamensis	0.207	0.158	0.160	0.175	0.179	0.002		
C. tucdupensis	0.206	0.149	0.140	0.171	0.166	0.140	0.006	

Table 3 Pairwise uncorrected p-distances based on 1,335 bp of ND2 and associated tRNAs calculated in MEGA v6.06 (Tamura et al., 2013) within the siamensis group.

Within species distances are presented in bold text and between species distances are presented below the diagonal.

	C. thachanaensis sp. nov.	C. phangngaensis sp. nov.	C. omari	C. chanardi	C. punctatonuchalis	C. vandeventeri	C. huaseesom	C. roticani	C. siamensis	
C. thachanaensis sp. nov.	0.003									
C. phangngaensis sp. nov.	0.260	0.002								
C. omari	0.282	0.112	0.042							
C. chanardi	0.243	0.114	0.118	–						
C. punctatonuchalis	0.211	0.250	0.264	0.255	–					
C. vandeventeri	0.144	0.252	0.266	0.240	0.211	–				
C. huaseesom	0.208	0.237	0.282	0.262	0.170	0.201	0.004			
C. roticani	0.275	0.088	0.090	0.117	0.256	0.268	0.274	0.002		
C. siamensis	0.134	0.250	0.278	0.255	0.194	0.123	0.194	0.281	–	

Systematics

Cnemaspis lineogularis sp. nov.	
urn:lsid:zoobank.org:act:8E3B21A4-93BF-4D08-B8D1-0A3EEF6BE44F	
Common name: Striped Throated Rock Gecko	
(Figs. 3–5)	

Holotype. BYU 62535 adult male, collected near Wat Khao Daeng, Kui Buri, Prachuap Khiri Khan, Thiland (12.134620°N, 99.961078°E; 12 m a.s.l.), 31 July 2016, by PLW, LLG, CA, MC, MSG, MLM.

Paratopotypes. BYU 62536 adult male and ZMKU R 00728 adult female paratypes bear the same collection and data as the holotype.

Figure 3 (A) male holotype BYU 62535 and (B) female paratype ZMKU R 00728 of Cnemaspis lineogularis sp. nov.

Diagnosis. Cnemaspis lineogularis is distinguished from all other species of Cnemaspis in the chanthaburiensis group by the combination of the following morphological and color pattern characters: maximum SVL 38 mm; nine supralabials; eight infralabials; ventral scales smooth; no precloacal pores; 13 paravertebral tubercles linearly arranged; no tubercles on the lower flanks; lateral caudal furrows present; no caudal tubercles in the lateral furrows; ventrolateral caudal tubercles anteriorly; caudal tubercles not encircling tail; subcaudals smooth bearing a single median row of enlarged smooth scales; lateral caudal tubercle row absent; shield-like subtibial scales absent; one post cloacal tubercle in males; no enlarged femoral or submetatarsal scales; enlarged femoral scales; subtibials smooth; 27–29 subdigital fourth toe lamellae; sexually dimorphic for dorsal color pattern; gular region yellow-orange, thick, black lineate markings in males, absent in females; subcaudal region whitish (Tables 4–6).

Description of the holotype. Adult male; SVL 38 mm; head oblong in dorsal profile, moderate in size (HL/SVL 0.25), somewhat narrow (HW/SVL 0.16), flattened (HD/HL 0.38), head distinct from neck; snout moderate (ES/HL 0.52), snout slightly concave in lateral view; postnasal region concave medially; scales on rostrum smooth becoming keeled posteriorly, raised, larger than conical scales on occiput; weak to absent supra ocular ridges; frontalrostralis sulcus deep; canthus rostralis nearly absent, smoothly rounded; eye large (ED/HL 0.26); extra-brillar, fringe scales largest anteriorly; pupil round; ear opening more round than oval; rostral slightly concave, dorsal 80% divided by longitudinal mediangroove; rostral bordered posteriorly by supra nasals and one small azygous scale and laterally by first supralabials; 9,9 (R,L) slightly raised supralabials decreasing in size posteriorly; 8,8 (R,L) infralabials decreasing in size posteriorly; nostrils elliptical, oriented dorsoposteriorly; bordered by small postnasal scales; mental large, triangular, concave, bordered posteriorly by three postmentals; gular and throat scales raised, smooth, small and round.

Figure 4 Ventral coloration and sexual dichromatism in the type series of Cnemaspis lineogularis sp. nov.

(A) adult male holotype BYU 62535, (B) adult male paratype BYU 62536, (C) adult female paratype ZMKU R 00728.

Figure 5 Dorsal view of the type series of Cnemaspis lineogularis sp. nov.

(A) adult male holotype BYU 62535, (B) adult male paratype BYU 62536, (C) adult female paratype ZMKU R 00728.

Table 4 Mensural and meristic character states for the type series of Cnemaspis lineogularis sp. nov.

All measurements taken are in millimeters and the abbreviations are defined in the materials and methods.

	BYU 62535 Holotype	BYU 62536 Paratype	ZMKU R 00728 Paratype	
Supralabials	9	9	9	
Infralabials	8	8	8	
Ventral scales keeled (1) or smooth (0)	0	0	0	
No. of precloacal pores	0	0	0	
Precloacal pores continuous (1) or separated (0)	/	/	/	
Precloacal pores elongate (1) or round (0)	/	/	/	
No. of paravertebral tubercles	13	13	13	
Tubercles linearly arranged (1) or more random (0)	1	1	1	
Tubercles present (1) or absent (0) on lower flanks	0	0	0	
Lateral caudal furrows present (1) or absent (0)	1	/	1	
Caudal tubercles in lateral furrow (1) or not (0)	0	/	0	
Ventrolateral caudal tubercles anteriorly (1) or not (0)	1	/	1	
Lateral caudal tubercle row present (1) or absent (0)	1	/	1	
Caudal tubercles restricted to a single				
paravertebral row on each side (1) or not (0)	0	/	1	
Subcaudals keeled (1) or smooth (0)	0	/	0	
Single median row of keeled subcaudals (1) or smooth (0)	0	/	0	
Caudal tubercles encircle tail (1) or not (0)	0	/	0	
Enlarged median subcaudal scale row (1) or not (0)	1	/	1	
No. of postcloacal tubercles in males	1	1	/	
Enlarged femoral scales present (1) or absent (0)	1	1	1	
Shield-like subtibial scales present (1) or absent (0)	0	0	0	
Subtibial scales keeled (1) or smooth (0)	0	0	0	
Enlarged submetatarsal scales on 1st toe (1) or not (0)	0	0	0	
No. of 4th toe lamellae	29	27	29	
SVL	38	35	29	
TL	48	b	24	
TW	2.9	/	2.66	
FL	6.5	6.3	4.34	
TBL	7.3	7.1	5.27	
AG	17.5	15.7	10.6	
HL	9.6	9.6	5.95	
HW	6.4	6.3	5	
HD	3.7	3.56	3.1	
ED	2.5	1.78	1.71	
EE	2.68	2.9	2.19	
ES	5	4.2	3.54	
EN	3.7	3.2	2.88	
IO	3.1	2.5	1.99	
EL	0.7	0.3	0.4	
IN	1.2	1.2	0.89	
Sex	m	m	f	
Notes.

m male

f female

/ data unavailable or absent

b broken

Body slender, elongate (AG/SVL 0.46); small, keeled, dorsal scales equal in size throughout body, intermixed with several large, multicarinate conical tubercles more or less randomly arranged; tubercles extend from the occiput to base of the tail; no tubercles on flanks; pectoral and abdominal scales smooth, not larger posteriorly; abdominal scales slightly larger than dorsals; no pore-bearing, precloacal scales or precloacal depressions; forelimbs moderately long, slender; dorsal scales slightly raised, multicarinate; ventral scales of brachia smooth, raised, juxtaposed; scales beneath forearm smooth, raised, subimbricate; palmar scales smooth, juxtaposed, raised; digits long with an inflected joint; claws recurved; sub digital lamellae unnotched; lamellae beneath first phalanges granular proximally, widened distally; lamellae beneath phalanx immediately following inflection granular, lamellae of distal phalanges wide; interdigital webbing absent; fingers increase in length from first to fourth with fourth and fifth equal in length; hind limbs slightly longer and thicker than forelimbs; dorsal scales raised, multicarinate, juxtaposed; dorsal scales on anterior margin of thighs enlarged, multicariante, becoming smaller posteriorly; ventral scales of thigh smooth; subtibial scales smooth, flat, imbricate, with no enlarged anterior row; plantar scales smooth, juxtaposed, raised; no enlarged submetatarsal scales beneath first metatarsal; digits elongate with an inflected jointed; claws recurved; subdigital lamellae unnotched; lamellae beneath first phalanges granular proximally, widened distally; lamellae beneath phalanx immediately following inflection granular, lamellae of distal phalanges wide; interdigital webbing absent; toes increase in length from first to fourth with fourth and fifth equal in length; 29,28 (R,L) subdigital lamellae on fourth toe; caudal scales similar to dorsal scale size, enlarge caudal tubercles arranged in segmented whorls, no encircling tail; caudal scales keeled, juxtaposed anteriorly; shallow, middorsal furrow; deeper, single, lateral furrow; enlarged, median, subcaudal scales; subcaudals smooth; median row of enlarged, keeled, subcaudal scales; transverse, tubercle rows do not encircle tail; caudal tubercles absent from lateral furrow; 2,1 (R,L) enlarged, postcloacal tubercles on lateral surface of hemipenal swellings at base of tail; posterior 30% of tail regenerated.

Table 5 Diagnostic color pattern characters separating various species of Cnemaspis from one another following Grismer et al., 2014d.

	chanthaburiensis group	siamensis group	
	lineogularis sp. nov.	chanthaburiensis	neangthyi	laoensis*	aurantiacopes	caudanivea	nuicamensis	tucdupensis	phangngaensis sp. nov.	thachanaensis sp. nov.	siamensis	huaseesom	chanardi	omari	roticanai	punctatonuchalis	vandeventeri	kamolnorranathi*	
Dorsal color pattern sexually dimorphic	yes	no	no	/	yes	no	no	no	yes	yes	no	yes	no	no	yes	yes	no	no	
Ventral pattern sexually dimorphic	yes	yes	no	/	yes	no	no	no	yes	yes	yes	yes	yes	/	yes	yes	yes	/	
Head yellow	no	no	no	no	no	no	no	no	no	no	no	yes	no	no	no	no	no	no	
Reddish blotches on head and body	no	no	no	no	no	no	no	no	no	no	no	no	no	no	no	no	no	no	
Dense yellow reticulum on occiput and side of neck	no	no	no	no	no	no	no	no	no	no	no	no	no	no	no	no	no	no	
Ocelli on occiput and nape	no	no	no	no	no	no	no	no	no	no	no	no	no	no	no	no	no	no	
Ocelli on shoulder	no	no	no	no	no	no	no	no	no	no	no	no	no	no	no	no	no	no	
Dual ocelli on shoulder	no	no	no	no	no	no	no	no	no	no	no	no	no	no	no	no	no	no	
Ocelli on brachium and side of neck	no	no	no	no	no	no	no	no	no	no	no	no	no	no	no	yes	no	no	
Thin, white, nuchal loop	no	no	no	no	no	no	no	no	no	no	no	no	no	no	no	no	no	no	
Large, black round spots on nape and anterior of body	no	no	no	no	no	yes	no	no	no	no	no	no	no	no	no	no	no	no	
Thin yellow reticulum on side of neck	no	no	no	no	no	no	no	no	no	no	no	no	no	no	no	no	no	no	
Yellowish, prescapular crescent	no	no	no	no	no	no	no	no	yes?	no	no	no	yes	yes	yes	no	yes	var	
Forelimbs yellow	no	no	no	no	no	no	no	no	no	no	no	yes	no	no	no	no	no	no	
Hind limbs yellow	no	no	no	no	no	no	no	no	no	no	no	yes	no	no	no	no	no	no	
Reddish blotches or bands on limbs	no	no	no	no	no	no	no	no	no	no	no	no	no	no	no	no	no	no	
Forearms and forelegs orange	no	no	no	no	yes	no	no	no	no	no	no	no	no	no	no	no	no	no	
Dorsal ground color magenta	no	no	no	no	no	no	no	no	no	no	no	no	no	no	no	no	no	no	
Dorsal ground color reddish	no	no	no	no	no	no	no	no	no	no	no	no	no	no	no	no	no	no	
Uniform brown ground color	no	no	no	no	no	no	no	no	no	no	no	no	no	no	no	no	no	no	
Light vertebral stripe	no	no	no	no	no	no	no	no	no	no	no	no	no	no	no	no	no	no	
Yellow postscapular band	no	no	no	no	no	no	no	no	no	no	no	no	no	no	no	no	no	no	
Black, squarish, paired, paravertebral dorsal markings	no	no	no	no	no	no	no	yes	no	no	no	no	no	no	no	no	no	no	
Small, light, round spots on flanks	no	no	no	no	no	no	no	no	no	no	no	no	no	no	no	no	no	no	
Black flanks with distinct yellowish spots	no	no	no	no	no	no	no	no	no	no	no	no	no	no	no	no	no	no	
Yellow or white bars on flanks	no	no	no	no	no	no	no	no	yes	no	no	no	no	no	no	no	no	no	
Original tail yellow	no	no	no	no	no	no	no	no	no	no	no	var	no	no	no	no	no	no	
Original tail orange	no	no	no	no	yes	no	no	no	no	no	no	no	no	no	no	no	no	no	
Regenerated tail yellow	no	no	no	/	no	/	no	no	/	no	no	no	no	no	yes	/	no	no	
Regenerated tail orange	no	no	no	/	no	/	no	no	/	no	no	no	no	no	no	/	no	no	
White, dorsal caudal tubercles	no	no	no	no	no	no	no	no	no	no	no	no	no	no	no	no	no	no	
Caudal bands present	no	yes	yes	yes	yes	yes	yes	yes	yes	yes	yes	yes	yes	yes	yes	yes	yes	yes	
Wide black and yellow bands on tail	no	no	no	no	no	yes	no	yes	no	no	no	no	no	no	no	no	no	no	
Thin, yellow caudal bands anteriorly	no	no	no	no	no	no	no	no	no	no	no	no	no	no	no	no	no	no	
Posterior portion of original tail white	no	no	no	no	no	yes	no	no	no	no	no	no	no	no	no	no	no	no	
Posterior portion of original tail black	no	no	no	no	no	no	no	yes	/	no	no	no	no	no	no	no	no	no	
Disticnt black and white caudal bands at least posteriorly	no	no	no	no	no	no	no	no	/	no	no	no	no	no	no	no	no	no	
Gular region orange	yes	yes	no	/	yes	no	no	yes	no	yes	no	no	no	no	no	/	yes	no	
Gular region yellow	no	no	no	/	no	no	no	no	yes	no	yes	yes	yes	yes	yes	/	no	no	
Lineate gular markings	yes	no	no	/	yes	no	yes	no	no	yes	yes	no	no	no	no	/	no	no	
Throat yellow	no	no	no	/	no	no	no	no	no	no	yes	yes	no	no	no	no	no	no	
Throat orange	no	yes	no	/	yes	no	no	yes	no	yes	no	no	no	no	no	yes	no	no	
Pectoral region yellow	no	no	no	/	no	no	no	no	no	no	yes	yes	no	no	yes	/	no	no	
Pectoral region orange	no	no	no	/	yes	no	no	yes	no	no	no	no	no	no	no	/	var	no	
Abdomen yellow	no	no	no	/	no	no	no	no	yes	no	no	no	yes	yes	yes	/	no	no	
Abdomen orange	no	yes	no	/	yes	no	no	yes	no	no	no	no	no	no	no	/	yes	no	
Ventral surfaces of forelimbs orange	no	no	no	/	yes	no	no	yes	no	no	no	no	no	no	no	/	yes	no	
Ventral surfaces of forelimbs yellow	no	no	no	/	no	no	no	no	no	no	no	yes	no	no	no	/	no	no	
Ventral surfaces of hind limbs orange	no	no	no	/	yes	no	no	yes	no	no	no	no	no	no	no	/	yes	no	
Ventral surfaces of hind limbs yellow	no	no	no	/	no	no	no	no	no	no	no	no	yes	yes	yes	/	no	no	
Subcaudal region yellow	no	no	no	/	no	no	no	no	yes	no	no	yes	yes	yes	yes	no	no	no	
Subcaudal region orange	no	yes	no	/	no	no	yes	no	no	no	no	no	no	no	no	yes	yes	no	
At least posterior half of subcaudal region white	no	no	no	no	no	yes	no	no	no	no	no	no	no	no	no	no	no	no	
Notes.

/ data unavailable

* indicate species that are not included in the molecular analyses

Table 6 Diagnostic morphological characters separating C. lineogularis from species of Cnemaspis in the chanthaburiensis group.

	lineogularis sp. nov.	chanthaburiensis	neangthyi	laoensis*	aurantiacopes	caudanivea	nuicamensis	tucdupensis	
Maximum SVL	38	42.2	54.0	40.9	58.4	47.2	48.2	51.0	
Supralabials	9	8–10	11–13	9	9–11	8,9	7–9	8–10	
Infralabials	8	7–10	10–12	7	8–10	7,8	6–7	7–9	
Ventral scales keeled (1) or smooth (0)	0	0	0	0	0	0	0	0	
No. of precloacal pores	0	6–9	2	/	0	0–2	3–6	0	
Precloacal pores continuous (1) or separated (0)	/	1	1	/	/	0	0	/	
Precloacal pores elongate (1) or round (0)	/	0,1	0	/	/	0,1	0,1	/	
No. of paravertebral tubercles	13	21–25	20–26	22	23–31	20–24	16–21	16–22	
Tubercles linearly arranged (1) or more random (0)	1	1	1	0	1	1	1	w,1	
Tubercles present (1) or absent (0) on lower flanks	0	1	1	1	1	0	1	1	
Lateral caudal furrows present (1) or absent (0)	1	1	1	1	1	1	1	1	
Caudal tubercles in lateral furrow (1) or not (0)	0	1	1	1	0	0, ant	0, ant	0	
Ventrolateral caudal tubercles anteriorly (1) or not (0)	1	0	1	0	1	1	1	1	
Lateral caudal tubercle row present (1) or absent (0)	1	1	1	0	1	0, ant	0, ant	0	
Caudal tubercles restricted to a single paravertebral row on each side (1) or not (0)	0	0	0	0	0	0	0	0	
Subcaudals keeled (1) or smooth (0)	0	0	0	0	0	0	0	0	
Single median row of keeled subcaudals (1) or smooth (0)	0	0	0	0	0	0	0	0	
Caudal tubercles encircle tail (1) or not (0)	0	0	0	0	0	0	0	0	
Enlarged median subcaudal scale row (1) or not (0)	1	0, post, w	1	w	1	0,w	1	w	
No. of postcloacal tubercles in males	1	1–3	1	2,3	1,2	1,2	2–4	0–3	
Enlarged femoral scales present (1) or absent (0)	1	0	0	0	0	0	0	0	
Shield–like subtibial scales present (1) or absent (0)	0	0	0	0	0	1	0	0	
Subtibial scales keeled (1) or smooth (0)	0	0,1	1	1	1	0,w	0	0	
Enlarged submetatarsal scales on 1st toe (1) or not (0)	0	0	0	0	1	0,w	0	1	
No. of 4th toe lamellae	27–29	22–29	22–25	29	27–31	23–30	27–33	26–32	
Sample size (n =)	3	8	5	1	17	9	10	11	
Notes.

w weak

ant anterior

post posterior

* species that are not included in the molecular analyses

/ data unavailable or absent

Character abbreviations follow those of Grismer et al. (2014d).

Coloration. In life, dorsal ground color of head light beige-green, that of the body, limbs and tail slightly lighter than head; top of the head bearing, small black and light green markings; thin diffuse broken dark-brown to black postorbital stripe, extending to the nape; two dark lines radiating distally from orbit; dark paravertebral markings extend from nape to anterior fourth of tail where they transform into diffuse incomplete bands, intermixed with sage colored paravertebral blotches; single dark prescapular blotch dorsoanteriorly of forelimb insertion; limbs slightly lighter than dorsal ground color with randomly placed, diffuse dark blotches; all ventral surfaces grayish white, except gular region and anteriormost portion of throat orange with black midgular stripe and adjacent black stripes along the mandibular margin; posterior margin of orange gular coloration edged with black, transverse markings (Figs. 3–5).

Figure 6 (A) habitat and (B) microhabitat of Cnemaspis lineogularis sp. nov.

Variation. Paratypes approximate the holotype (BYU 62535) in general aspects of coloration except that the female paratype (ZMKU R 00728) lacks the black markings in the gular region and the yellowish-orange gular coloration is less prominent, additionally the dorsal coloration is much lighter. Selected body measurements and variation in squamation are presented in Table 4.

Etymology. The specific epithet lineogularis is derived from the Latin adjective linues for the word “line” and the nominative form of the Latin word gulare meaning “throat” and is in reference to the multiple dark gular lines present in the males of this species.

Distribution. Only known from the type locality but we hypothesize it will be found in nearby karst formations (Figs. 1 and 6).

Natural history. The type series and several other individuals were active during the day in shaded areas and would rapidly retreat to nearby cracks and crevices at the slightest provocation. We hypothesize this may be due to high predation as we found Trimeresurus fucatus in an ambush posture in the same microhabitat. No individuals were seen deep within the caves and from our observations, it appears this species primarily inhabits the more exterior surfaces of the karst tower (Fig. 6). The karst formations in this area are extensive and we assume this species has a much wider distribution than that reported here. We hypothesize that diurnality in this species is to avoid competition with and predation from the much larger Cyrtodactylus somroiyot with which it is hypothesized to be syntopic with. This is a commonly observed pattern among syntopic pairs of Cnemaspis and Cyrtodactylus throughout their distributions in Southeast Asia (Grismer et al., 2014d, and references therein).

Comparisons. Cnemaspis lineogularis sp. nov. can be differentiated for all other species in the chanthaburiensis group based on the following morphological and color pattern characteristics (see Tables 5 and 6 for additional comparisons). Cnemaspis lineogularis sp. nov. differs from C. chanataburiensis, C. neangthyi, C. laoensis, C. aurantiacopes, C. caudanivea, C. nuicamensis, and C. tucdupensis by having a smaller maximum SVL (38 mm vs. 42.2 mm, 54.0 mm, 40.9 mm, 58.4 mm, 47.2 mm, 48.2 mm, and 51.0 mm, respectively), by having less paravertebral tubercles (13 vs. 21–25, 20–26, 22, 23–31, 20–24, 16–21, and 16–22 respectively), and by having enlarged femoral scales. Cnemaspis lineogularis sp. nov. is further differentiated from C. neangthyi by having less supralabial scales (9 vs. 11–13). Cnemaspis lineogularis sp. nov. differs from C. neangthyi by having less infralabial scales (8 vs. 10–12) and from C. nuicamensis by having more infralabial scales (8 vs. 6–7). It is further differentiated from C. chanthaburiensis, C. neangthyi, C. aurantiacopes, C. caudanivea, and C. nuicamensis by lacking precloacal pores. From C. loaensis, C. lineogularis sp. nov. differs by having linearly arranged tubercles versus randomly arranged tubercles. Cnemaspis lineogularis sp. nov. differs from C. chanthaburiensis, C. neangthyi, C. laoensis, C. aurantiacopes, C. nuicamensis, and C. tucdupensis by lacking tubercles on the lower flanks. Cnemaspis lineogularis sp. nov. differs from C. chanthaburiensis, C. neangthyi, C. laoensis, by lacking caudal tubercles in the lateral furrow. Cnemaspis lineogularis sp. nov. has ventrolateral caudal tubercles anteriorly which separates it from C. chanthaburiensis and C. laoensis which lack this character. Cnemaspis lineogularis sp. nov. differs from C. loaensis, C. caudanivea, C. nuicamensis, and C. tucdupensis by the presence of a lateral caudal tubercle row. From C. chanthaburiensis, C. laoensis, C. caudanivea, and C. tucdupensis, C. lineogularis sp. nov. differs by having an enlarged median subcaudal scale row. C. lineogularis sp. nov. differs from C. laoensis and C. nuicamensies by having one postcloacal tubercle in males versus 2,3 and 2–4 respectively. C. lineogularis sp. nov. is further differentiated from C. caudanivea by lacking shield-like subtibial scales. Cnemaspis lineogularis sp. nov. differs from C. neangthyi, C. laoensis, and C. aurantiacopes by lacking keeled subtibial scales. Cnemaspis lineogularis sp. nov. differs from C. aurantiacopes and C. tucdupensis by lacking an enlarged submetatarsal scale on the 1st toe. Cnemaspis lineogularis sp. nov. is further differentiated from from C. neangthyi by having more 4th toe lamellae (27–29 vs. 22–25). Cnemaspis lineogularis sp. nov. is further differentiated from all other species in the chanthaburiensis group based on squamation and color pattern characteristics (Tables 5 and 6).

Cnemaspis phangngaensis sp. nov.	
urn:lsid:zoobank.org:act:6053C709-A409-4F65-B15C-8C647D7EDF1C	
Common name: The Phangnga Rock Gecko	
(Figs. 7–9)	

Holotype. BYU 62538 adult male, collected at Phung Chang Cave, Mueang Phangnga District, Phangnga Province, Thailand (8.442344°N, 98.514869°E; 12 m a.s.l.), 26 July 2016, by PLW, LLG, CA, MC, MSG, MLM.

Figure 7 (A) adult male holotype BYU 62538 and (B) female paratype BYU 62537 of Cnemaspis phangngaensis sp. nov.

Figure 8 Ventral coloration and sexual dichromatism of Cnemaspis phangngaensis sp. nov.

(A) male holotype BYU 62538 and (B) female paratype BYU 62537.

Paratopotype. BYU 62537 adult female paratype bears all the same collection and locality information as the holotype.

Diagnosis. Cnemaspis phangngaensis sp. nov. is distinguished from all other species of Cnemaspis in the siamensis group by the combination of the following morphological and color pattern characteristics: maximum SVL 42 mm; 10 supralabials; 10 infralabials; ventral scales keeled; four continuous precloacal scales bearing a single round pore in males; 22 paravertebral tubercles linearly arranged; no tubercles on the lower flanks; lateral caudal furrows present; no caudal tubercles in the lateral furrows; lateral caudal tubercle row present; ventrolateral caudal tubercles anteriorly; caudal tubercles not encircling tail; caudal tubercles restricted to a single paravertebral row; subcaudals keeled bearing a single median row of enlarged keeled scales; two post cloacal tubercle in males; no enlarged femoral, tibial, or sub metatarsal scales; subtibials keeled; no enlarged median subcaudal scale row; no submetatarsal scale on first toe; 29 subdigital fourth toe lamellae; no enlarged median subcaudal scale row; dorsal and ventral color pattern sexually dimorphic; yellow or white bars present on flanks; prescapular marking present; anterior gular region dark yellowish, no dark lineate markings in males or females, and no mid-gular marking; posterior gular region and pectoral region whitish in males; abdomen yellow; subcaudal region yellow (Tables 5–7).

Figure 9 Dorsal coloration of the type series of Cnemaspis phangngaensis sp. nov.

(A) male holotype BYU 62538 and (B) female paratype BYU 62537.

Description of the holotype. Adult male; SVL 42 mm; head oblong in dorsal profile, moderate in size (HL/SVL 0.27), somewhat narrow (HW/SVL 0.16), flattened (HD/HL 0.35), head distinct from neck; snout moderate (ES/HL 0.44), slightly concave in lateral view; postnasal region concave medially; scales on rostrum smooth becoming keeled posteriorly, raised, larger than conical scales on occiput; weak to absent supra ocular ridges; frontalrostralis sulcus shallow; canthus rostralis nearly absent, smoothly rounded; eye large (ED/HL 0.20); extra-brillar, fringe scales largest anteriorly; pupil round; ear opening more oval, taller than wide; rostral slightly concave, dorsal 80% divided by longitudinal median groove; rostral bordered posteriorly by supra nasals and one small azygous scale and laterally by first supralabials; 10, 10 (R,L) slightly raised supralabials decreasing in size posteriorly; 10, 10 (R,L) infralabials decreasing in size posteriorly; nostrils elliptical, oriented dorsoposteriorly; bordered by small postnasal scales; mental large, triangular, concave, bordered posteriorly by three postmentals; gular and throat scales raised, keeled, small and round.

Body slender, elongate (AG/SVL 0.45); small, raised, keeled, dorsal scales equal in size throughout body, intermixed with several large , multicarinate conical tubercles more or less randomly arranged; tubercles extend from the occiput to base of the tail; no tubercles on flanks; pectoral and abdominal scales keeled, not larger posteriorly; abdominal scales slightly larger than dorsals; two pore-bearing, continuous, precloacal pores on each side; forelimbs moderately long, slender; dorsal scales slightly raised, keeled; ventral scales of brachia smooth, raised, juxtaposed; scales beneath forearm smooth, slightly raised, subimbricate; palmar scales smooth, juxtaposed, raised; digits long with an inflected joint; claws recurved; sub digital lamellae unnotched; lamellae beneath first phalanges granular proximally, widened distally; lamellae beneath phalanx immediately following inflection granular, lamellae of distal phalanges wide; interdigital webbing absent; fingers increase in length from first to fourth with fourth and fifth equal in length; hind limbs slightly longer and thicker than forelimbs; dorsal scales raised, multicarinate, juxtaposed; ventral scales of thigh, slightly raised, conical, keeled; subtibial scales keeled, flat, imbricate, with no enlarged anterior row; plantar scales smooth, juxtaposed, raised; no enlarged submetatarsal scales beneath first metatarsal; digits elongate with an inflected jointed; claws recurved; subdigital lamellae unnotched; lamellae beneath first phalanges granular proximally, widened distally; lamellae beneath phalanx immediately following inflection granular, lamellae of distal phalanges wide; interdigital webbing absent; toes increase in length from first to fourth with fourth and fifth equal in length; 29, 29 (R,L) subdigital lamellae on fourth toe; caudal scales similar to dorsal scale size, enlarge caudal tubercles arranged in segmented whorls, not encircling tail; caudal scales keeled, juxtaposed anteriorly; shallow, middorsal furrow; deeper, single, lateral furrow; enlarged, median, subcaudal scales; subcaudals keeled; median row of enlarged, keeled, subcaudal scales; transverse, tubercle rows do not encircle tail; caudal tubercles absent from lateral furrow; 1, 1 (R,L) enlarged flat, postcloacal tubercle on lateral surface of hemipenal swellings at base of tail; posterior ∼30% of tail missing.

Coloration. In life dorsal ground color of head light beige, that of the body, limbs and tail slightly darker than the head with darker irregular blotches; top of the head bearing, small black and sage markings; thin diffuse broken dark brown to black postorbital stripe, extending to the nape, not complete; light sage vertebral blotches extending form the nape to tail where they transform into diffuse near complete irregular bands; intermixed with light sage blotches; single light-yellowish prescapular crescent dorsoanteriorly of forelimb insertion; flanks with irregular incomplete sage to yellowish-orange bars becoming more orange distally; limbs slightly darker than dorsal ground color with randomly placed, diffuse dark and sage colored blotches; all ventral surfaces grayish-white, except gular, abdominal, and subcaudal regions are yellowish-orange, with more pronounced darker yellow stippling (Figs. 7–9).

Table 7 Mensural and meristic character states for the type series of Cnemaspis phangngaensis sp. nov.

All measurements are taken in millimeters and the abbreviations are defined in the materials and methods.

	BYU 62538 holotype	BYU 62537 paratype	
Supralabials	10	10	
Infralabials	10	10	
Ventral scales keeled (1) or smooth (0)	1	1	
No. of precloacal pores	4	0	
Precloacal pores continuous (1) or separated (0)	1	/	
Precloacal pores elongate (1) or round (0)	0	/	
No. of paravertebral tubercles	22	22	
Tubercles linearly arranged (1) or more random (0)	1	1	
Tubercles present (1) or absent (0) on lower flanks	0	0	
Lateral caudal furrows present (1) or absent (0)	1	1	
Caudal tubercles in lateral furrow (1) or not (0)	0	0	
Ventrolateral caudal tubercles anteriorly (1) or not (0)	1	1	
Lateral caudal tubercle row present (1) or absent (0)	1	1	
Caudal tubercles restricted to a single paravertebral row on each side (1) or not (0)	1	1	
Subcaudals keeled (1) or smooth (0)	1	1	
Single median row of keeled subcaudals (1) or smooth (0)	1	1	
Caudal tubercles encircle tail (1) or not (0)	0	0	
Enlarged median subcaudal scale row (1) or not (0)	0	0	
No. of postcloacal tubercles in males	2	/	
Enlarged femoral scales present (1) or absent (0)	0	0	
Shield-like subtibial scales present (1) or absent (0)	0	0	
Subtibial scales keeled (1) or smooth (0)	1	1	
Enlarged submetatarsal scales on 1st toe (1) or not (0)	0	0	
No. of 4th toe lamellae	29	30	
SVL	42	41	
TL	23b	44	
TW	3.3	3.2	
FL	6.37	6.6	
TBL	8.26	8.23	
AG	19.28	17.6	
HL	11.6	11.1	
HW	6.79	6.56	
HD	4.1	4.1	
ED	2.4	2.4	
EE	3.1	3.1	
ES	5.17	4.86	
EN	3.9	4.2	
IO	2.6	2.9	
EL	1	0.99	
IN	2.85	2.75	
Sex	m	f	
Notes.

m male

f female

/ data unavailable or absent

b broken

Variation in the type series. The female paratype (BYU 62537) approximates the holotype in general aspects of coloration except the overall dorsal coloration is lighter and the ventral coloration is a uniform light yellow and is not as prominent in the gular and abdominal regions. Select body measurements and variation in squamation are presented in Table 7.

Etymology. The specific epithet phangngaensis is a noun in apposition to the type locality where this species is found.

Distribution. Only known from the karst formation in which it is found, the Phung Chang Cave, Phangnga, Mueang Phangnga, Thailand. We hypothesize that this species will be found on nearby contiguous karst formations.

Natural history. Cnemaspis phangngaensis inhabits a karst formation in a lowland limestone forest (Fig. 10) surrounded by highly disturbed, urbanized habitat. The male holotype was collected at night on the karst approximately 15 m above the ground on the exterior surface of the tower and the female was collected at night sleeping on a leaf approximately 1.2 m above the limestone forest floor adjacent to the nearby karst formation. Individuals were also observed active during the day, but avoided being caprtured be retreating into the rock crevices. We hypothesize that these are diurnal karst dwellers that use the vegetation at night for refuge. We hypothesize that diurnality in this species is to avoid competition with and predation from the much larger Cyrtodactylus lekaguli with which it is syntopic.

Figure 10 (A) general karst and limestone forest near the type locality of Cnemaspis phangngaensis sp. nov. (B) karst microhabitat where C. phangngaensis occurs.

Table 8 Diagnostic morphological characters separating species of Cnemaspis from one another in the siamensis group.

	phangngaensis sp. nov.	thachanaensis sp. nov.	siamensis	huaseesom	chanardi	omari	roticanai	punctatonuchalis	vandeventeri	kamolnorranathi*	
Maximum SVL	42	39	39.7	43.5	40.1	41.3	47.0	49.6	44.7	37.8	
Supralabials	10	10,11	8,9	7–10	7–10	8,9	8,9	8	8,9	8,9	
Infralabials	10	9–11	6–8	6–9	6–8	7,8	7,8	7,8	7–9	7,8	
Ventral scales keeled (1) or smooth (0)	1	1	1	0	1	1	1	0	1	0,w	
No. of precloacal pores	4	0	0	5–8	6–8	4	3–6	0	4	7	
Precloacal pores continuous (1) or separated (0)	1	/	/	1	0	0	0	/	0	1	
Precloacal pores elongate (1) or round (0)	0	/	/	0	0	0	0	/	0	1	
No. of paravertebral tubercles	22	15–19	19–25	18–24	20–30	22–29	25–27	24–27	25–29	19–24	
Tubercles linearly arranged (1) or more random (0)	1	1	0	w,0	0	w,0	0	w	0	w	
Tubercles present (1) or absent (0) on lower flanks	0	1	1	1	1	w,1	1	1	0	1	
Lateral caudal furrows present (1) or absent (0)	1	1	1	1	1	1	1	1	1	1	
Caudal tubercles in lateral furrow (1) or not (0)	0	0	0	1	0	0	1	0	0	1	
Ventrolateral caudal tubercles anteriorly (1) or not (0)	1	1	0	0	0	0	0	1	0	0	
Lateral caudal tubercle row present (1) or absent (0)	1	1	1	0	1	1	0	ant	1	1	
Caudal tubercles restricted to a single paravertebral row on each side (1) or not (0)	1	1	0	0	0	0	0	0	0	0	
Subcaudals keeled (1) or smooth (0)	1	1	1	0	1	1	1	0	1	1	
Single median row of keeled subcaudals (1) or smooth (0)	1	1	0	0	0	0	0	0	w	w	
Caudal tubercles encircle tail (1) or not (0)	0	0	0	0	0	1	0	0	0	0	
Enlarged median subcaudal scale row (1) or not (0)	0	0	1	0	1	0	w	1	1	w	
No. of postcloacal tubercles in males	2	0	1,2	1,2	1	1	1,2	1–3	1–3	1,2	
Enlarged femoral scales present (1) or absent (0)	0	0	0	0	0	0	0	0	0	0	
Shield–like subtibial scales present (1) or absent (0)	0	0	0	0	0	0	0	0	0	0	
Subtibial scales keeled (1) or smooth (0)	1	1	1	0	1	1	1	0	1	0,1	
Enlarged submetatarsal scales on 1st toe (1) or not (0)	0	1	0	0	0	0	0	0	0	0	
No. of 4th toe lamellae	29	24	24–26	21–31	25–30	25–28	26–29	29–31	24–28	24–28	
Sample size	2	6	12	5	25	4	8	5	3	3	
Notes.

w weak

ant anterior

post posterior

* species that are not included in the molecular analyses

/ data unavailable or absent

Character abbreviations follow that of Grismer et al. (2014d).

Comparisons. The phylogenetic analysis recovers the chanardi group and C. phangngaensis sp. nov. as the sister species to a clade containing C. omari and C. roticani (Fig. 2). This relationship is further supported by the following derived morphological characters (sensu Grismer et al., 2014d), prescapular crescent present, yellow abdomen, yellow ventral surfaces of the hind limbs and tail being yellow and numerous other morphological and color pattern characteristics (Tables 5 and 8). C. phangngaensis sp. nov. differs from C. chanardi, C. omari, and C. roticanai by having; more infralabial scales (10 vs. 6–8, 7,8, and 7,8, respectively); continuous precloacal pores; paravertebral tubercles linearly arranged; lacking tubercles on the lower flank; ventrolateral caudal tubercles anteriorly; caudal tubercles restricted to a single paraveterbral row on each side; a single median row of keeled subcaudals. Cnemaspis phangngaensis sp. nov. is further differentiated from C. chanardi and C. omari by having a larger maximum SVL (42 mm vs. 40.1 mm and 41.3 mm, respectively). Cnemaspis phangngaensis sp. nov. differs from C. omari, and roticani by having more supralabial scales (10 vs. 8,9 and 8,9, respectively). C. phangngaensis sp. nov. differs from C. chanardi by having fewer precloacal pores (4 vs. 6–8). Cnemaspis phangngaensis sp. nov. differs from C. roticani by having fewer paravertebral tubercles (22 vs. 25–27). From C. roticanai, C. phangngaensis sp. nov. differs by lacking caudal tubercles in the lateral furrow and by having a lateral caudal tubercle row present. Cnemaspis phangngaensis sp. nov. differs from C. omari by lacking caudal tubercles encircling the tail and by having more lamellae under the 4th toe (29 vs. 25–28). Cnemaspis phangngaensis sp. nov. is further differentiated from C. chanardi by lacking an enlarged median subcaudal scale row. From C. chanardi and C. omari, C. phangngaensis differs by have two postcloacal tubercles in males versus one. Cnemaspis phangngaensis is further differentiated from all other species in the siamensis group based on squamation and color pattern characteristics (Tables 5 and 8).

Cnemaspis thachanaensis sp. nov.	
urn:lsid:zoobank.org:act:3581C94E-6170-4F42-9159-E2B564B576F1	
Common name: The Tha Chana Rock Gecko	
(Figs. 11–13)	
Cnemaspis kamolnorranathi (Grismer et al., 2010, pg. 29)	
Cnemaspis kamolnorranathi (Grismer et al., 2014d, pg. 130)	

Holotype. BYU 62544 adult male, collected at Tham Khao Sonk hill, Tha Chana District, Surat Thani Province, Thailand (9.549878°N, 99.175544°E; 107 m a.s.l.), 30 July 2016, by PLW, LLG, CA, MC, MSG, MLM.

Paratopotypes. All paratypes (BYU 62542–62543, ZMKU R 00729–00731) bear the same collection and locality data as the holotype.

Diagnosis. Cnemaspis thachanaensis sp. nov. is distinguished from all other species of Cnemaspis in the siamensis group by the combination of the following morphological and color pattern characteristics: maximum SVL 39 mm; 10 or 11 supralabials; 9–11 infralabials; ventral scales keeled; no precloacal pores in males; 15–19 paravertebral tubercles linearly arranged; tubercles generally present on the lower flanks; lateral caudal furrows present; no caudal tubercles in the lateral furrows; ventrolateral caudal tubercles anteriorly; presence of lateral caudal tubercle row; caudal tubercles not encircling tail; caudal tubercles restricted to a single paravertebral row; subcaudals keeled bearing a single median row of enlarged keeled scales; one or two post cloacal tubercles in males; no enlarged femoral or tibial scales; subtibials keeled; enlarged submetatarsal scale on first toe; 23–25 subdigital fourth toe lamellae; sexually dimorphic for ventral and dorsal coloration; yellow or white bars present on flanks; prescapular marking present; gular region yellowish-orange, dark incomplete lineate markings in males, less prominent in females; abdomen, limbs and subcaudal region whitish (Table 9).

Figure 11 Coloration of Cnemaspis thachanaensis sp. nov.

(A) male holotype BYU 62544 and (B) BYU 62542 female paratype.

Figure 12 Ventral coloration and sexual dichromatism of the type series of Cnemaspis thachanaensis sp. nov., males: (A) BYU 62543, (B) BYU 62544 (holotype), (C) ZMKU R 00731, females: (D) ZMKU R 00729, (E) ZMKU R 00730, (F) BYU 62542.

Figure 13 Dorsal coloration of the type series of Cnemaspis thachanaensis sp. nov., males: (A) BYU 62543, (B) BYU 62544 (holotype), (C) ZMKU R 00731, females: (D) ZMKU R 00729, (E) ZMKU R 00730, (F) BYU 62542.

Description of the holotype. Adult male; SVL 33 mm; head oblong in dorsal profile, moderate in size (HL/SVL 0.29), somewhat narrow (HW/SVL 0.16), flattened (HD/HL 0.37), head distinct from neck; snout moderate (ES/HL 0.44), snout slightly concave in lateral view; postnasal region concave medially; scales on rostrum smooth becoming keeled posteriorly, raised, larger than conical scales on occiput; weak to absent supra ocular ridges; frontalrostralis sulcus shallow; canthus rostralis nearly absent, smoothly rounded; eye large (ED/HL 0.22); extra-brillar, fringe scales largest anteriorly; pupil round; ear opening more oval than round, taller than wide; rostral slightly concave, dorsal 80% divided by longitudinal median groove; rostral bordered posteriorly by supra nasals and one small azygous scale and laterally by first supralabials; 11, 11 (R,L) slightly raised supralabials decreasing in size posteriorly; 10, 10 (R,L) infralabials decreasing in size posteriorly; nostrils elliptical, oriented dorsoposteriorly; bordered by small postnasal scales; mental large, triangular, concave, bordered posteriorly by three postmentals; gular scales small, smooth, raised and round; throat scales subimbricate, keeled, small and round.

Table 9 Menusural and meristic character state for the type series of Cnemaspis thachanaensis sp. nov.

All measurements are taken in millimeters and the abbreviations are defined in the materials and the methods.

	BYU	ZMKU R	BYU	BYU	ZMKU R	ZMKU R	
	62544	00731	62543	62542	00729	00730	
	holotype	paratype	paratype	paratype	paratype	paratype	
Supralabials	10	11	10	10	10	10	
Infralabials	10	11	10	10	9	9	
Ventral scales keeled (1) or smooth (0)	1	1	1	1	1	1	
No. of precloacal pores	0	0	0	/	/	/	
Precloacal pores continuous (1) or separated (0)	/	/	/	/	/	/	
Precloacal pores elongate (1) or round (0)	/	/	/	/	/	/	
No. of paravertebral tubercles	15	19	15	17	15	16	
Tubercles linearly arranged (1) or more random (0)	1	1	1	1	1	1	
Tubercles present (1) or absent (0) on lower flanks	1	1	0	1	1	1	
Lateral caudal furrows present (1) or absent (0)	1	1	1	1	1	1	
Caudal tubercles in lateral furrow (1) or not (0)	0	0	0	0	0	0	
Ventrolateral caudal tubercles anteriorly (1) or not (0)	1	1	1	1	1	1	
Lateral caudal tubercle row present (1) or absent (0)	1	1	1	1	1	1	
Caudal tubercles restricted to a single paravertebral row on each side (1) or not (0)	1	1	1	1	1	1	
Subcaudals keeled (1) or smooth (0)	1	1	1	1	1	1	
Single median row of keeled subcaudals (1) or smooth (0)	1	1	1	1	1	1	
Caudal tubercles encircle tail (1) or not (0)	0	0	0	0	0	0	
Enlarged median subcaudal scale row (1) or not (0)	0	0	0	0	0	0	
No. of postcloacal tubercles in males	0	0	0	/	/	/	
Enlarged femoral scales present (1) or absent (0)	1	1	1	0	0	0	
Shield-like subtibial scales present (1) or absent (0)	0	0	0	0	0	0	
Subtibial scales keeled (1) or smooth (0)	1	1	1	1	1	1	
Enlarged submetatarsal scales on 1st toe (1) or not (0)	1	1	1	1	1	1	
No. of 4th toe lamellae	24	23	24	25	23	23	
SVL	33	37	34	39	35	35	
TL	41	44r	43	46	6b	16b	
TW	3.6	3.91	3.32	4	3.4	3.7	
FL	5.97	5.68	5.39	5.83	5	5.9	
TBL	6.8	7.3	6.19	7.14	6.9	6.5	
AG	14.55	16	13.1	17.72	14.49	15.6	
HL	9.79	10.3	9.74	11.49	9.37	9.8	
HW	5.55	6.27	5.3	6.6	5.45	5.7	
HD	3.64	4.1	3.8	4.32	3.9	3.8	
ED	2.21	2.4	2	2.36	2.04	1.9	
EE	2.85	2.79	2.6	2.79	2.67	3	
ES	4.4	4	3.6	4.29	4.29	4.57	
EN	2.5	1.18	2.6	3.1	2.79	3.7	
IO	2.58	2.97	2.3	3.29	3.1	2.8	
EL	0.76	0.82	0.67	1	0.71	0.75	
IN	2.1	2.8	2.4	2.2	2.29	2.2	
Sex	m	m	m	f	f	f	
Notes.

m Male

f Female

/ Data unavailable or absent

b Broken

r regenerated

Body slender, elongate (AG/SVL 0.44); small, raised, keeled, dorsal scales equal in size throughout body, intermixed with several large, multicarinate conical tubercles more or less randomly arranged; tubercles extend from the occiput to base of the tail; enlargerd multicarinate conical tubercles on flanks; pectoral and abdominal scales keeled, not larger posteriorly; abdominal scales slightly larger than dorsals; no pore-bearing, precloacal pores on either side; forelimbs moderately long, slender; dorsal scales slightly raised, keeled; ventral scales of brachia smooth, raised, juxtaposed; scales beneath forearm smooth, slightly raised, subimbricate; palmar scales smooth, juxtaposed, raised; digits long with an inflected joint; claws recurved; sub digital lamellae unnotched; lamellae beneath first phalanges granular proximally, widened distally; lamellae beneath phalanx immediately following inflection granular, lamellae of distal phalanges wide; interdigital webbing absent; fingers increase in length from first to fourth with fourth and fifth equal in length; hind limbs slightly longer and thicker than forelimbs; dorsal scales raised, multicarinate, juxtaposed; ventral scales of thigh, slightly raised, conical, keeled; subtibial scales keeled, flat, imbricate, with no enlarged anterior row; plantar scales smooth, juxtaposed, raised; enlarged submetatarsal scales beneath first metatarsal; digits elongate with an inflected jointed; claws recurved; subdigital lamellae unnotched; lamellae beneath first phalanges granular proximally, widened distally; lamellae beneath phalanx immediately following inflection granular, lamellae of distal phalanges wide; interdigital webbing absent; toes increase in length from first to fourth with fourth and fifth equal in length; 24,24 (R,L) subdigital lamellae on fourth toe; caudal scales similar to dorsal scale size, enlarge caudal tubercles arranged in segmented whorls, not encircling tail; caudal scales keeled, juxtaposed anteriorly; shallow, middorsal furrow; deeper, single, lateral furrow; enlarged, median, subcaudal scales; subcaudals keeled; median row of enlarged, keeled, subcaudal scales on last 2/3 of tail; transverse, tubercle rows do not encircle tail; caudal tubercles absent from lateral furrow; 1,1 (R,L) enlarged flat, postcloacal tubercle on lateral surface of hemipenal swellings at base of tail.

Coloration. In life dorsal ground color of head light-brown, that of the body, limbs and tail slightly darker than the head with even darker irregular blotches; top of the head bearing, small dark-brown and light-green markings; thin diffuse broken dark brown to black postorbital stripe, extending to the nape, not complete; light-green vertebral blotches extending form the nape to tail where they transform into diffuse near complete irregular bands intermixed with dark brown blotches turning into bands posteriorly; flanks with irregular incomplete small light-green colored blotches to yellow-orange bars becoming smaller posterior; limbs much lighter than dorsal ground color, limbs grayish-white and dark brown incomplete irregular bands; all ventral surfaces grayish-white, except gular and throat regions are yellow-orange not restricted to the gular region and extend onto the throat and the anterior region of the pectoral region in males, incomplete transverse markings in the gular region in male and is less prominent in females (Figs. 11–13).

Variation. The paratypes approximate the holotype (BYU 62544) in general aspects of morphology except that the female paratypes lack precloacal pores and yellow-orange gular regions. Paratypes ZMKU R 00731, BYU 62542, and BYU 62541 have more paravertebral tubercles (19, 17, 16 respectively vs. 15), dark irregular gular spots not as prominent in females (Fig. 12). Select body measurements and additional variation in squamation are presented in Table 9.

Etymology. The specific epithet thachanaensis is a noun in apposition to the type locality where this species is found.

Distribution. This species is only know from the type locality Thom Sonk Hill, Tha Chana District, Surat Thani Province, Thailand and we expect that it will be found on nearby adjacent karst formations (Fig. 14).

Figure 14 (A) karst and limestone forest near the type locality of Cnemaspis thachanaensis sp. nov. (B) karst microhabitat of Cnemaspis thachanaensis sp. nov.

Natural history. Cnemaspis thachanaensis inhabits a karst tower embedded within a highly disturbed lowland limestone forest. One male individual was observed during the day situated upside down on a karst overhang displaying its yellow-orange throat by doing push-ups. All other specimens were found active during the day on the karst and we hypothesize that these are diurnal karst dwellers. No specimens were observed at night. Grismer et al. (2010) noted that one specimen (CUMZ-R 2009,624-3) was collected on a vine near the adjacent limestone. Karst dwelling species of Cnemaspis have been know to sleep on vegetation at night (Grismer et al., 2010; Grismer et al., 2014d, P Wood, pers. obs., 2016). This species may use the vegetation at night for refuge to avoid Cyrtodactylus thirakaputhi which is nocturnal and maybe a potential predator.

Remarks. Specimen CUMZ-R 2009,6,24-3 was collected from Thom Sonk Hill, Tha Chana District, Surat Thani Province and was described as C. kamolnorranathi in Grismer et al. (2010). Grismer et al. (2010) noted that the relatively wide separation (∼110 km) between the type locality of C. kamolnorranathi (Petchphanomwat Waterfall, Tai Rom Yen National Park, Ban Nasan District, Surat Thai Province) and the locality of the paratype CUMZ-R 2009,6,24-3 from Thom Sonk Hill, Tha Chana District, Surat Thani Province and suggested that there are probably undiscovered, geographically intervening populations in the appropriate habitat separating these two localities (Grismer et al., 2014d). Grismer et al. (2010) and Grismer et al. (2014d) noted that there is exceptional intrapopulational variation in the degree of keeling of the ventral and the subtibial scales in C. kamolnorranathi suggesting the possibility that C. kamolnorranathi may be composed of multiple species. After examining additional specimens from Thom Sonk Hill, Tha Chana District, Surat Thani Province (BYU 62542, ZMKU R 00729–00731 and the paratype CUMZ-R 2009,6,24-3) we determined that CUMZ-R 2009,6,24-3 is not conspecific with C. kamolnorranathi and with additional specimens it can be diagnosed as a new species (see comparisons below for details). Here we remove CUMZ-R 2009,6,24-3 from C. kamolnorranthi and place it in C. thachanaensis sp. nov. restricting C. kamolnorranathi to the Petchphanomwat Waterfall, Tai Rom Yen National Park, Ban Nasan District, Surat Thai Province. There are no genetic samples of C. kamolnorranthi available to further test this hypotheses, however we present strong morphological evidence separating these species.

Comparisons. Cnemaspis thachanaensis sp. nov. is the sister species to a clade containing C. siamensis and C. vandeventeri (Fig. 2). Although we were not able to obtain genetic material for C. kamolnorranathi we compare it here using morphology to demonstrate that the paratype (CUMZ-R 2009,6,24-3, MS101) is conspecific with C. thachanaensis sp. nov. Cnemaspis thachanaensis sp. nov. differs from C. siamensis and C. vandeventeri by having a smaller SVL (39 mm, vs. 39.7 mm and 44.7 mm) and by having a larger maximum SVL from C. kamolnorranathi (39 mm vs. 37.8 mm). C. thachanaensis sp. nov. differs from C. siamensis, C. vandeventeri, and C. kamolnorranathi by; having more supralabial scales (10–11 vs. 8–9, 8–9, 8–9, respectively); having more infralabials (9–11 vs. 6–8, 7–9, and 7–8, respectively); having paravertebral tubercles linearly arranged; having ventrolateral caudal tubercles anteriorly; having caudal tubercles restricted to a single paravertebral row on each side; having a single median row of keeled subcaudal scales; lacking a single enlarged subcaudal scale row; lacking postcloaclal tubercles in males; the presence of an enlarged submetatarsal scale on the 1st toe. Cnemaspis thachanaensis sp. nov. is further differentiated from C. kamolnorranathi by having keeled ventral scales. Cnemaspis thachanaensis sp. nov. differs from C. vandeventeri and C. kamolnorranathi by lacking precloacal pores. We can further differentiate C. thachanaensis sp. nov. from C. vandeventeri by having less paravertebral tubercles (15–19 vs. 25–29). Cnemaspis thachanaensis sp. nov. differs from C. kamolnorranathi by lacking tubercles in the lateral furrow. Cnemaspis thachanaensis sp. nov. is further differentiated the more distantly related species C. huaseesom and C. punctatonuchalis in the siamensis group by having a smaller maximum SVL (39 mm vs. 43.5 mm and 49.6 mm, respectively); having more supralabials 10,11 vs. 8; having caudal tubercles restricted to a single paravertebral row; having keeled ventral scales; single median row of keeled subcaudals; lacking enlarged median subcaudal scale row; by lacking postcloacal tubercles in males. Cnemaspis thachanaensis sp. nov. differs by having more infralabials 9–11 vs. 7, 8 in C. punctatonuchalis. Cnemaspis thachanaensis sp. nov. differs from C. huaseesom by lacking precloacal pores. From C. huaseesom, C. thachanaensis sp. nov. differs by having ventrolateral caudal tubercles anteriorly and the presence of a lateral caudal tubercle row. Cnemaspis thachanaensis sp. nov.differs from C. punctatonuchalis by having keeled subcaudal scales. Cnemaspis thachanaensis sp. nov. differs from C. huaseesom by having keeled subtibial scales an an enlarged submetatarsal scale on the first toe. From C. punctatonuchalis, C. thachanaensis sp. nov. differs by having less fourth toe lamellae, 24 vs. 29–31. Cnemaspis thachanaensis sp. nov. is differentiated from all other species in the siamensis group based on squamation and color pattern characteristics (Tables 5 and 8).

Discussion

The discovery of three new species of karst-dwelling Cnemaspis from Peninsular Thailand is not surprising, given the nature of the vastly unexplored karst and limestone forests dispersed throughout this area. Peninsular Malaysia received considerable attention with respect to herpetofaunal surveys, yet new karst-dwelling species are being discovered and described every year (see Grismer et al., 2016a, for a summary). The results of these surveys have resulted in the discovery of 14 species of geckos (including Cnemaspis and Cyrtodactylus as well as two snakes Grismer et al. (2016a)). In comparison, Peninsular Thailand has received little attention with most of the focus on the genus Cyrtodactylus resulting in the discovery and description of 15 species in the last 55 years, with 14 of these being described in the last 15 years (see Table 6 in Grismer et al., 2016a). However, there has been limited field research on the Thai karst-dwelling Cnemaspis from these areas (Grismer et al., 2010). With the small amount of time spent in Phangnga, Tha Chana, and Prachuap Khiri Khan, we were able to discover three new species (C. lineogularis sp. nov., C. phangngaensis sp. nov., C. thachanaensis sp. nov.) and successfully collect genetic samples of C. punctatonuchalis and C. vandeventeri. We expect that as more time is focused collecting specimens from the unexplored karst formations additional new species will be discovered.

From the fieldwork that has been conducted on the Thai and Malaysian karst formations a fair amount of Cyrtodactylus and Cnemaspis have been discovered and described. On some of these formations both the nocturnal Cyrtodactylus and diurnal Cnemaspis occur syntopically. For example both Cyrtodactylus lekaguli and Cnemaspis phangngaensis sp. nov. at Phung Chang Cave, Phangnga, Thailand, both Cyrtodactylus astrum and Cnemaspis omari in Perlis, Malaysia, and both Cyrtodactylus langkawiensis and Cnemaspis roticani on Pulau langkawi, Malaysia. Comparing the phylogenetic relationships of these Cyrtodactylus with the Cnemaspis reveals an identical phylogeographic pattern (Phangnga (Perlis, Pulau Langkawi)). Corroborating these relationships are the calculated average mean pairwise sequence divergence within the respective genera (Cyrtodactylus 9.4% (Grismer et al., 2016b) and Cnemaspis 9.6% [this study]). From these preliminary analyses we hypothesize that the formation of these karst formations may have been the resulting factor for simultaneous speciation events within these two respective genera. With additional fieldwork and data collection in these areas more detailed analyses with divergence times can be estimated to investigate the timing of these divergence events.

The inclusion of C. punctatonuchalis and C. vandeventeri in the phylogenetic analyses helps test previous morphological hypotheses set forth by Grismer et al. (2010) and Grismer et al. (2014d), and has also contributed towards a more complete phylogeny of the genus Cnemaspis (49 of the 55 named species including the three new species described herein). Cnemaspis punctatonuchalis was nested within the siamensis group confirming the placement solely based on morphological and color pattern characteristics by Grismer et al. (2014d), which was also hypothesized to be more closely related to the other northern species (north of the Isthmus of Kra, C. huaseesom and C. siamensis). This is further supported here as the sister species to C. huaseesom (Fig. 2). Cnemaspis vandeventeri was hypothesized based on its distribution that it should align with the siamensis group, however Grismer et al. (2014d) also suggested that the presence of a light prescapular crescent that diagnoses a monophyletic group composed of C. chanardi, C. phangngaensis, C. omari and C. roticanai may suggest that it is more closely related to this group. The phylogenetic placement of C. vandeventeri is well nested in the siamensis group confirming the placement based on its distribution of Grismer et al. (2010) and Grismer et al. (2014d), however the hypothesis that it may be more closely related to the group with the prescapular crescent is not supported by our phylogenetic hypothesis and could represent an instance of convergent evolution of the prescapular crescent. This in not surprising based on the well documented parallel/convergent evolution present in the genus Cnemaspis (Grismer et al., 2014d), and further analyses to address hypotheses pertaining to parallel/convergent evolution of multiple traits are in preparation (P Wood et al., 2016, unpublished data).

The phylogenetic position of C. lineogularis as the sister taxon to the entire chanthaburiensis group, indicates a trans-Gulf of Thailand relationship with other species from southern Indochina. This is not a novel biogeographic pattern and the close relationship between Indochinese and Malaysian lineages has been observed in Butterfly lizards in the genus Leiolepis (Grismer et al., 2014a), and in some species of Cyrtodactylus (Grismer et al., 2015). However, the previous documented cases of this pattern are much further south on the peninsula. This pattern could easily be explained by previous cyclic sea level fluctuations that exposed the Sunda Shelf providing multiple dispersal corridors between the Thai-Malay Peninsula and Indochina (e.g., Voris, 2000; Sathiamurthy & Voris, 2006; Woodruff, 2010). Further investigation into the biogeographic patterns for Cnemaspis are in preparation (P Wood et al., 2016, unpublished data), and with the continued discovery of new species of Cnemasapis in the area, these broader studies will contribute to the understanding of the complex biogeography patterns on the Thai-Malay Peninsula. The discovery of three new species of Cnemaspis described here underscores the need for additional fieldwork in the karst towers of the Thai-Malay Peninsula and the surrounding areas to aid in conservation efforts, document the herpetofauna diversity, and provide data for biogeographic studies.

Supplemental Information

Table S1 Specimens used for the molecular phylogenetic analyses.

Voucher number abbreviations are as follows: ABTC, Australian Biological Tissue Collection; AMS, Australian Museum, Sydney; ANWC, Australian National Wildlife Collection; BYU, Monte L. Bean Life Science Museum at Brigham Young University; CAS, California Academy of Sciences; FMNH, Field Museum of Natural History; HC, Herpetological Collection of the Universiti Kebangsaan Malaysia, Bangi, Selangor; ID, Indraneil Das field series; JB, Jon Boone captive collection; MVZ, Museum of Vertebrate Zoology (Berkeley); LSUHC, La Sierra University Herpetological Collection; LSUMZ, Louisiana State University Museum of Zoology; MZB, Museum Zoologicum Bogoriense, Cibinung, Java, Indonesia; RAH, Rod A. Hitchmough field series; TG, Tony Gamble; USMHC, Universiti Sains Malaysia Herpetological Collection at the Universiti Sains Malaysia, Penang, Malaysia; USNM, United States National Museum (Smithsonian); YPM, Yale Peabody Museum; ZMKU, Zoological Museum Kasetsart University, Thailand; ZRC, Zoological Reference Collection, Raffles Museum.

Click here for additional data file.

We are thankful for general discussions and logistical conversations with Attapol Rujirawan (aka Bank), Siriporn Yodthong, Natee Ampai, Korkhwan Termprayoon, Piyawan Phuanprapai, and Sengvilay Seateun. We would also like to thank Todd R. Jackman, Angelica Crottini, and D. James Harris for providing feedback that greatly improved the quality of the manuscript.

Additional Information and Declarations

Competing Interests

Author Contributions

Animal Ethics

Field Study Permissions

DNA Deposition

New Species Registration

The authors declare there are no competing interests.

Perry Lee Wood Jr conceived and designed the experiments, performed the experiments, analyzed the data, wrote the paper, prepared figures and/or tables, reviewed drafts of the paper.

L. Lee Grismer conceived and designed the experiments, performed the experiments, wrote the paper, reviewed drafts of the paper.

Anchalee Aowphol, César A. Aguilar, Micheal Cota, Marta S. Grismer and Matthew L. Murdoch performed the experiments, reviewed drafts of the paper.

Jack W. Sites Jr contributed reagents/materials/analysis tools, reviewed drafts of the paper.

The following information was supplied relating to ethical approvals (i.e., approving body and any reference numbers):

Brigham Young University’s Institutional Animal Care and Use Committee (IACUC) has approved the animal use protocol for this study (protocol # 160401).

The following information was supplied relating to field study approvals (i.e., approving body and any reference numbers):

We received a collecting permit from the National Research Council Thailand (NRCT) to PLW and AA (No. 0002/7606).

The following information was supplied regarding the deposition of DNA sequences:

GenBank accession numbers KY091231 –KY091244.

The following information was supplied regarding the registration of a newly described species:

Publication ID: urn:lsid:zoobank.org:author:E2AF9554-5062-48C6-84AA-FA50D720DEF7

Family: Gekkonidae, Genus: Cnemaspis, Species: lineogularis urn:lsid:zoobank.org:act:8E3B21A4-93BF-4D08-B8D1-0A3EEF6BE44F

Family: Gekkonidae, Genus: Cnemaspis, Species: phangngaensis urn:lsid:zoobank.org:act:6053C709-A409-4F65-B15C-8C647D7EDF1C

Family: Gekkonidae, Genus: Cnemaspis, Species: thachanaensis urn:lsid:zoobank.org:act:3581C94E-6170-4F42-9159-E2B564B576F1

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
