# Peer review of "Three new karst-dwelling Cnemaspis Strauch, 1887 (Squamata; Gekkoniade) from Peninsular Thailand and the phylogenetic placement of C. punctatonuchalis and C. vandeventeri"

_PeerJ, doi:10.7717/peerj.2884_

## Round 0.1 · original submission · Minor Revisions

Your paper has received three very positive reviews and all three reviewers have suggested some minor revisions for you to consider. I hope you find these reviews helpful and look forward to your revised work. Congratulations and thanks for submitting to PeerJ.

·

Basic reporting

No comments

Experimental design

No comments

Validity of the findings

No comments

Additional comments

This is an interesting article, describing three new species. I have no doubt that it deserves publication. Overall I have only minor suggestions for changes prior to publication.

Abstract and various places throughout: replace "paravetebral" with "paravertebral"
Abstract - give the mtDNA gene used after the sequence divergence values (at least the first time)
L21 and narrow
L24 geologically
Its not an essential change, but Modeltest has been superseded by jModeltest for some years now. I am sure it will make no difference to the results, but the authors should consider using the later version.
L123 "significantly supported" - I wouldnt use the word significant when referring to bootstraps. Change to "well supported" or something equally vague.
Regarding measurements, it is always preferable if they are made by the same person to avoid biases. I think its worth mentioning who made the measurements so that it is clear if this is the case.
L175 and 179 replace "bare" with "demonstrate"
L154 delete "the fact"
L190 and 392 Giving max SVLs as a character when you only have a couple of individuals measured seems limited to me, especially when the SVLs are 42 vs 40.1 and 41.3. I dont think it matters, as you have many diagnostic scalation characters but I would suggest giving max SVL less importance.
Sentences beginning L549 and L560: both are poorly constructed. I suggest rewriting them.

·

Basic reporting

Basic reporting: The article is well-written and clear. However, there are a few places where too much fore-knowledge is required to easily understand the article. This can easily be fixed by the addition of a few paragraphs. The keywords are too specific and should include words that would allow someone searching for taxonomy or Cnemaspis or day geckos could find this paper. In the introduction after an excellent discussion of karst regions of the Malay Peninsula, both Cyrtodactylus and Cnemaspis are discussed with no introduction or explanation.
(The abbreviation of MP is not necessary as it is only used 3 times in the paper – twice in the same paragraph that the abbreviation was defined in.)
Suggested added paragraph one in the introduction: a general discussion of Cnemaspis as a genus including species numbers, variation in diurnaity/nocturnality and variation in color pattern and habitat. Also, Cyrtodactylus is discussed without defining it intitially. Cyrtodactylus also needs to be introduced as a nocturnal karst-dwelling gecko that can co-occur with Cnemaspis and is a potential competitor. Adding few natural history comments about both genera would help as well.

Suggested paragraphs 2 and 3 (possibly 4) in the discussion: A discussion of the fact that these new species occur in clades where nocturnality and diurinality is variable and therefore potentially labile, making hypotheses of shifts in activity pattern discussed in the natural history section of the species descriptions possible. The fact that all three new species are apparently diurnal should appear earlier as well, either in the abstract and/or introduction.
I would also like to see some comment on the coloration of the new species relative to the SE Asian Cnemaspis as a whole. For example, are all members of the chantaburiensis and siamensis groups cryptic in dorsal coloration?
I think a discussion paragraph comparing karst Cnemaspis to karst Cyrtodactylus would help make this paper of more general interest and spark future hypotheses. Is the timing and degree of diversification of the two genera comparable in age and number of species? This can be answered somewhat qualitatively and would serve to act as a hypothesis that could be tested in the future using dating methods.

Small things in the manuscript:
What is the correct spelling? In the original species description and in the 2014 Grismer et al. paper, Cnemaspis chanthaburiensis is used consistently – In this paper chantaburiensis (with no “h” is used throughout the paper) – Which one is correct, and why the recent change?
L21 change from “an narrow” to “and narrow”

Experimental design

The experimental design I take to mean the scope of characters (both morphological and molecular) examined in order to provide a clear species diagnosis and description. As such, the experimental design is excellent, but see my comments regarding the presentation of the data below.

Validity of the findings

I think that the combination of molecular and morphological diagnostic features make it clear that these are clearly well-defined, distinct species.

The clarity of the finding can be improved by making the following changes:
Figure 1. Indicate the 3 new species described in the paper – either by boxing or circling the names, or a new symbol.
Figure legend “The asterisk (*) identifies a species… “ because there is only one.

Figure 2: A dot or dash should be used on the outline phylogeny to indicate the SE Asian Cnemaspis clade --- As presented, it looks like the chathaburiensis (or chatanburiensis) clade is the sister group to all other Cnemaspis, but that is not the case. Optionally, I think that indicating Nocturality/Diurnality would help to clarify a discussion of activity pattern (maybe Closed/Open circles).

Table 2 is only mentioned once in the paper in the results – It would be more meaningful to compare these divergences to closely related karst Cyrtodactylus, if there is a comparable small group of species.

Additional comments

I would like to see more herpetological taxonomic papers in PeerJ; this is an excellent paper, but I think that a more general audience should be kept in mind when introducing and discussing Cnemaspis.
I realize that discussing nocturnally/diurnality could take focus away from the main point, but I think a quick, qualitative discussion would help the paper.

·

Basic reporting

- I confirm that the manuscript is clear, provide unambigous data and it is written in professional English.
- The manuscript provide sufficient introduction and background information, with maybe the only exception of the systematic revision of the entire genus. As far as I know, the genus Cnemaspis still include species from Africa and India/Sri Lanka (although it is known that those groups do not form a monophyletic group). Therefore, the authors might consider to add one sentence on the manauscript on the topic, for the seek of clarity and to provide the reader with the most updated systematic knowledge on the group.
- This contribution is structured following PeerJ standards
- All figures are relevant and of high quality, well labelled and described
- genbank accession number still need to be provided, and potentially teh author can decide to submit their alignment in a public repository.

Experimental design

- I confirm this manuscript provide an original research within the scope of the journal
- The research question are well defined, relevant and meaningful and the authors clearly state how research fills an identified knowledge gap.
- I confirm the manuscript follows a rigorous investigation performed to a high technical & ethical standard.
- I confirm that the methods are described with sufficient detail and information so that could be easily replicated

Validity of the findings

- The manuscript provide robust data to confirm teh validity of the three newly described species
- Data are extremely robust and nicely described.
- The conclusion of the work are very well justified and are linked to original research question (filling the gap of knowledge on this group of geckos)
- no speculation are provided

Additional comments

I left several minor suggestions for change in the .pdf version of your contribution.

---

## Round 0.2 · accepted · Accept

Thank you for your careful revision of your exciting manuscript on three new geckos. I have reviewed your revisions and except for a few typos which I've noted to the PeerJ staff, it looks great. I am happy to recommend acceptance.